

# A chromosome-scale genome assembly of mungbean (*Vigna radiata*)

Supaporn Khanbo[1], Poompat Phadphon[1], Chaiwat Naktang[1], Duangjai Sangsrakru[1], Pitchaporn Waiyamitra[1], Nattapol Narong[1], Chutintorn Yundaeng[1], Sithichoke Tangphatsornruang[1], Kularb Laosatit[2], Prakit Somta[2] and Wirulda Pootakham[1]

[1] National Omics Center, National Center for Genetic Engineering and Biotechnology (BIOTEC), National Science and Technology Development Agency (NSTDA), Pathum Thani, Thailand
[2] Department of Agronomy, Faculty of Agriculture at Kamphaeng Saen, Kasetsart University, Kamphaeng Saen Campus, Nakhon Pathom, Thailand

## ABSTRACT

**Background**. Mungbean (*Vigna radiata*) is one of the most socio-economically important leguminous food crops of Asia and a rich source of dietary protein and micronutrients. Understanding its genetic makeup is crucial for genetic improvement and cultivar development.

**Methods**. In this study, we combined single-tube long-fragment reads (stLFR) sequencing technology with high-throughput chromosome conformation capture (Hi-C) technique to obtain a chromosome-level assembly of *V. radiata* cultivar 'KUML4'.

**Results**. The final assembly of the *V. radiata* genome was 468.08 Mb in size, with a scaffold N50 of 40.75 Mb. This assembly comprised 11 pseudomolecules, covering 96.94% of the estimated genome size. The genome contained 253.85 Mb (54.76%) of repetitive sequences and 27,667 protein-coding genes. Our gene prediction recovered 98.3% of the highly conserved orthologs based on Benchmarking Universal Single-Copy Orthologs (BUSCO) analysis. Comparative analyses using sequence data from single-copy orthologous genes indicated that *V. radiata* diverged from *V. mungo* approximately 4.17 million years ago. Moreover, gene family analysis revealed that major gene families associated with defense responses were significantly expanded in *V. radiata*.

**Conclusion**. Our chromosome-scale genome assembly of *V. radiata* cultivar KUML4 will provide a valuable genomic resource, supporting genetic improvement and molecular breeding. This data will also be valuable for future comparative genomics studies among legume species.

## INTRODUCTION

Mungbean (*Vigna radiata* (L.) R. Wilczek var. *radiata*) is a socio-economically important legume crop with high nutritional value. It is a self-pollinated and fast-growing diploid plant species with 2n = 2x = 22 chromosomes (*Kang et al., 2014*). This legume crop belongs to the papilionoid subfamily of the Fabaceae and is widely cultivated in South and Southeast Asian countries, including India, China, Indonesia, Thailand, Myanmar, and the

Corresponding authors
Prakit Somta, agrpks@ku.ac.th
Wirulda Pootakham,
wirulda@alumni.stanford.edu

Philippines (*Sokolkova et al., 2020*). Globally, mungbean cultivation covers approximately 7.3 million hectares, with an average yield of 721 kilograms per hectare. India and Myanmar each contribute 30% to the total production of 5.3 million tons (*Nair & Schreinemachers, 2020*). This crop is an inexpensive source of dietary protein, carbohydrates, vitamins, and contains higher levels of folate and iron than most other legumes (*Keatinge et al., 2011*; *Liu et al., 2022*). The seeds of mungbean contain a high amount of protein and carbohydrates, comprising approximately 20–30% protein and 60–70% carbohydrates (*Somta et al., 2022*). Mungbean is popularly grown in crop rotation systems due to its short duration, relative drought tolerance, and ability to improve soil fertility through atmospheric nitrogen fixation in symbiosis with *Rhizobium* and *Bradyrhizobium* bacteria in the soil (*Somta et al., 2022*; *Yimram, Somta & Srinives, 2009*).

Reference genomes are key to understand genome organization, to describe evolutionary events and to precisely identify genomic regions/genes associated with agronomically important traits. With advances in DNA sequencing technologies, mungbean genomes have been sequenced and assembled. The first draft genome sequence of mungbean was published in 2014, with the initial assembly of the elite breeding line 'VC1973A' from Illumina short-read sequences (*Kang et al., 2014*). However, this draft assembly covered 80% (421 Mb) of the estimated genome size and consists of 2,748 scaffolds with an N50 length of 1.52 Mb. An improved mungbean genome of VC1973A using long sequence reads was then published (*Ha et al., 2021*). The genome assembly covered a total sequence of 476 Mb with an N50 length of 5.2 Mb. Recently, *Liu et al. (2022)* reported a high-quality reference genome assembly of the mungbean cultivar 'Jilv7', utilizing the PacBio Sequel II platform and Illumina paired-end (PE) read technology. The genome assembly was 479.35 Mb in size, with a contig N50 length of 10.34 Mb. The line VC1973A, developed by the World Vegetable Center (formerly the Asian Vegetable Research and Development Center) in 1982, has since been widely grown as elite cultivar in Korea, Thailand, Taiwan, Canada, and China (*Kang et al., 2014*). Different cultivars/genotypes within the same species are likely to have variations in genome content, structure, and gene numbers, making a single reference genome insufficient to represent the full spectrum of genomic diversity within the species (*Editorial, 2020*; *Golicz, Batley & Edwards, 2016*; *Saxena, Edwards & Varshney, 2014*). Having more than one reference genome for mungbean offers significant advantages for genetics improvement and breeding.

In this study, we employed single-tube long-fragment reads (stLFR) sequencing (*Wang et al., 2019*) combined with high-throughput chromosome conformation capture (Hi-C) technique (*Putnam et al., 2016*) to assemble and construct a chromosome-level genome assembly of the mungbean 'KUML4', a cultivar recently released in Thailand (*Somta et al., 2024*). We specifically chose this cultivar due to its agronomically desirable traits, including high yield, large seed size, early and uniform maturity, and resistance to leaf spot and powdery mildew diseases (*Somta et al., 2024*). The reference assembly provides valuable resource for genetic improvement in mungbean variety breeding programs. This genomic data will also be useful for future comparative genomic studies of legume species.
## MATERIALS & METHODS

### Sample collection and DNA extraction

We selected the mungbean cultivar KUML4 for genome sequencing and assembly. Fresh leaves were collected from a 4-week-old mungbean plant. The leaf samples were immediately frozen in liquid nitrogen and stored until use. High molecular weight (HMW) DNA was extracted from the frozen tissues using the Genomic-tip 100/G DNA preparation kit (Qiagen, Hilden, Germany) according to the manufacturer's instructions. After extraction, the concentration, integrity, and purity of the DNA were determined using the dsDNA HS assay on a Qubit fluorometer (Thermo Fisher Scientific, Waltham, USA) and pulsed-field gel electrophoresis (PFGE).

### stLFR library construction and sequencing

The stLFR sequencing library was constructed using a total of 10 ng of HMW DNA following the MGIEasy stLFR Library Prep Kit's instructions (MGI Tech, Shenzhen, China). In brief, the HMW DNA was first inserted with transposons. The transposon-inserted DNA was then incubated with clonal barcoded beads and subjected to random priming. Following that, the captured DNA molecules were disrupted to yield sub-fragments, each less than 1 kb in size. Adapters were ligated to the DNA fragments, and the library was developed by PCR. Finally, the library was sequenced on the DNBSEQ-G400 using the MGISEQ-2000RS Sequencing Flow Cell v3.0 (MGI Tech, Shenzhen, China).

### Hi-C library preparation and sequencing

A high-throughput chromosome conformation capture (Hi-C) technique (*Putnam et al., 2016*) was performed by Biomarker Technologies Corporation (Beijing, China) to construct the chromosome-level assembly of mungbean. Briefly, the chromatin was crosslinked with fresh formaldehyde, and the fixed chromatin was then digested with *HindIII* restriction endonuclease. Subsequently, the digested DNA and the overhanging 5′ ends of the DNA fragments were repaired with biotinylated nucleotides. DNA fragments were then circularized by blunt-end ligation. After the cross-linking was reversed, the DNA was purified and sheared into smaller fragments of 300–700 bp in size, which were then pulled down with streptavidin beads. The purified fragments were used to generate a sequencing library that was then sequenced on the Illumina platform (PE150; Illumina, San Diego, CA, USA).

### Genome assembly and Hi-C scaffolding

The stLFR reads were used to construct the preliminary draft genome assembly following the single-tube long fragment read data analysis software stLFRdenovo version 1.0.5, with default settings (https://github.com/BGI-biotools/stLFRdenovo/releases/tag/v1.0.5, accessed on 12 January 2024). This preliminary assembly was further scaffolded into a chromosome-level assembly using the high-throughput chromosome conformation capture (Hi-C) technique. The preliminary draft genome assembly and Hi-C reads were used as the input data for HiRise version 2.1.9, a software pipeline designed specifically for using proximity ligation data to scaffold genome assemblies (*Putnam et al., 2016*).

The Hi-C sequences were mapped to the draft input assembly using BWA version 0.7.17 (*Li & Durbin, 2009*) (https://github.com/lh3/bwa). HiC-Pro version 2.10.0 (*Servant et al., 2015*) was then applied for data filtering and quality assessment of the Hi-C reads. The separations of Hi-C read pairs mapped within draft scaffolds were analyzed by HiRise to produce a likelihood model for genomic distance between read pairs, and the model was used to identify and break putative misjoins, score prospective joins, and make joins above a threshold. Manual adjustment and curation were carried out using a Hi-C visualization tool to correct potential misplacements or orientation errors. The mungbean genome assembly was deposited in National Center for Biotechnology Information (NCBI) under the accession number JAVCZA000000000. The raw reads stLFR data were deposited in the SRA database at the NCBI with the accession number SRR29729411.

## Genome size estimation

To estimate the genome size, k-mer analysis was performed on the high-quality paired-end (PE) read distribution using Jellyfish software version 2.2.10, and the k-mer distribution was plotted with GenomeScope version 2.0 ($k = 21$; http://genomescope.org/genomescope2.0/; latest accessed: September 18, 2024) (*Vurture et al., 2017*).

## Genome assembly evaluation

After assembly, we evaluated the quality of the final genome assembly by aligning stLFR reads using BWA version 0.7.17 (*Li & Durbin, 2009*). We also used Benchmarking Universal Single-Copy Orthologs (BUSCO) version 5.4.4 (*Manni et al., 2021*) to assess the completeness of the assembly. The BUSCO pipeline was used to test for the presence and completeness of orthologs using the plant-specific database from Embryophyta OrthoDB release 10 (*Kriventseva et al., 2015*). Additionally, the LTR Assembly Index (LAI) was computed and assessed using LTR_retriever version 3.0.1 (*Ou & Jiang, 2018*).

## Repetitive sequence identification

The repetitive sequences in the assembled genome were identified with de novo and homology-based methods. First, we constructed a de novo repeat library of the mungbean using the software package RepeatModeler version 2.0.1 with default parameters (https://github.com/Dfam-consortium/RepeatModeler, accessed on 30 April 2024). This package includes distinct discovery algorithms, RECON version 1.08 and RepeatScout version 1.0.5 (*Bao & Eddy, 2002*; *Flynn et al., 2020*; *Price, Jones & Pevzner, 2005*). The programs were employed to identify the boundaries of repetitive elements and to build consensus models of interspersed repeats. After we obtained the repeat library, we aligned the repeat sequences to NCBI GenBank's non-redundant protein database using BLASTX with an e-value threshold of $10^{-6}$ to verify that the library did not contain sequences belonging to large families of protein-coding sequences. Then, using the RepBase plant repeat database (*Jurka et al., 2005*) (https://www.girinst.org/), homology-based repetitive sequences of the mungbean genome were identified with RepeatMasker version 4.0.9_p2 with default parameters (*Tempel, 2012*). Additionally, we employed Tandem Repeats Finder (TRF) version 4.09 (*Benson, 1999*) with default parameters to predict tandem repeats element. HelitronScanner version 1.0 (*Xiong et al., 2014*) and miteFinder version 3.0

(*Hu, Zheng & Shang, 2018*) were used to identify helitrons and miniature inverted-repeat transposable elements (MITEs), respectively.

## Gene annotation

Regarding gene annotation, we employed the BRAKER2 version 2.1.6 (*Brůna et al., 2021*) which performs gene prediction by using GeneMark-ES/ET/EP/ETP version 4.65 and AUGUSTUS version 3.4.0. BRAKER2 version 2.1.6 also enables the integration of transcript and protein homology information into the predictions. Six sets of RNA-Seq data from various tissues, SRS652317 (roots, stems, leaves), SRS1308070 (seed), SRR27190622 (petal blooming flower), SRS3960651 (V6-stage leaves), SRS9581523 (Thai SUT1 4-days-old root) and SRS702243 (5-day seedling hypocotyl) were retrieved from SRA database of NCBI. The RNA-Seq data were mapped onto the *V. radiata* assembly of this study with HiSAT2 version 2.2.1 (*Kim et al., 2019*) and used for BRAKER2 analysis. Protein data from Viridiplantae OrthoDB11 (*Kuznetsov et al., 2023*) and *V. radiata* cultivar VC1973A (GCF_000741045.1) were included in the analysis. The predicted genes were filtered and the longest isoforms of overlapping transcripts were selected using TSEBRA version 1.0.3 (*Gabriel et al., 2021*) with the default evidence-weight setting. BUSCO version 5.4.4 (*Manni et al., 2021*) was used with eudicots_odb10 to evaluate the quality of the annotation. Functional annotation of the predicted genes was performed using OmicsBox version 1.3.11 (https://www.biobam.com/download-omicsbox/, last accessed on 10 May 2024). We aligned the protein sequences against the UniProtKB/Swiss-Prot (*Bairoch & Apweiler, 2000*) and GenBank non-redundant (NR) databases *via* BLASTP (*Gish & States, 1993*) with an e-value cutoff of $10^{-5}$. Gene ontology (GO) terms were retrieved and assigned to *V. radiata* query sequences, while enzyme codes (EC) corresponding to *V. radiata* gene ontology were extracted and mapped to Kyoto Encyclopedia of Genes and Genomes (KEGG) pathway annotations. We also predicted tRNAs using the tRNAscan-SE tool version 2.0 (*Chan Patricia et al., 2021*) (http://lowelab.ucsc.edu/tRNAscan-SE/). As rRNAs are highly conserved, we used rRNA sequences from closely related species as references and used BLAST to predict rRNA sequences. Other ncRNAs, including miRNAs and snRNAs, were predicted by searching against the Rfam version 14.1 database (*Griffiths-Jones et al., 2005*) (http://infernal.janelia.org/) using the Infernal software version 1.1.5 with the default parameters. The genome annotation of mungbean was deposited in the Figshare database (https://figshare.com/articles/dataset/Vigna_radiata_annotation_files/27094231).

## Comparative genomics and phylogenetic analyses

OrthoFinder version 2.4.0 (*Emms & Kelly, 2019*) was used to identify orthologous gene groups among the three mungbean cultivars (KUML4, Vrad_JL7, and VC1973A v2) and 14 other plant species (Table S1). These included nine legumes [black gram (*Vigna mungo* (L.) Hepper), créole bean (*Vigna reflexo-pilosa* Hayata), cowpea (*Vigna unguiculata* (L.) Walp.), adzuki bean (*Vigna angularis* (Ohwi) Ohwi and Ohashi), common bean (*Phaseolus vulgaris* L.), barrel medic (*Medicago truncatula* Gaertn.), soybean (*Glycine max* (L.) Merr.), chickpea (*Cicer arietinum*) and peanut (*Arachis duranensis*)], three cucurbit species (cucumber (*Cucumis sativus* L.), melon (*Cucumis melo* L.) and watermelon (*Citrullus lanatus* L.)),

one rosid species (Arabidopsis), and one monocot species (rice (*Oryza sativa* L.)). The three mungbean cultivars were further clustered and subjected to Venn diagram analysis to explore the common and cultivar-specific genes across the three mungbean genomes. Single-copy orthologous protein sequences were aligned with MUSCLE version 3.8.1551 (*Edgar, 2004*), and alignment gaps were removed using trimAl version 1.4 rev15 (*Capella-Gutiérrez, Silla-Martínez & Gabaldón, 2009*). Alignment blocks were concatenated using catsequences program (https://github.com/ChrisCreevey/catsequences, last accessed on 22 May 2024), and the best substitution model for each block was estimated using ModelTest-NG program version 0.1.7 (*Darriba et al., 2019*). A maximum-likelihood phylogenetic tree was constructed using RAxML-NG program version 1.0.2 (*Kozlov et al., 2019*) with the best-fit substitution model, and 1,000 bootstrap replicates. *O. sativa* was designated as the outgroup of the phylogenetic tree. Divergence times of species within the phylogenetic tree were estimated with the MCMCTree program version 4.0 in the Phylogenetic Analysis by Maximum Likelihood software (PAML4 package) (*Yang, 2007*) using the relaxed-clock model with the known divergence time between cucumber and melon (8.4–11.8 million years ago; *Sebastian et al., 2010*). Finally, we employed CAFE software version 4.2 (*Han et al., 2013*) to predict the significant expansion and contraction of gene families across the phylogeny ($p$-value < 0.05) with the gene birth-death ($\lambda$) parameters estimated using the maximum-likelihood method. The expanded and contracted genes were then subjected to GO and KEGG functional annotation. Genome collinearity analysis was performed using the MCScan tool (Python version, https://github.com/tanghaibao/jcvi/) from jcvi v1.3.885 (*Tang et al., 2024*; *Wang et al., 2012*). MCscan was employed to define syntenic blocks between the *V. radiata* KUML4, *V. radiata* JL7, and *V. mungo* genomes. Collinearity between the three genomes was visualized using collinearity plots.

## RESULTS

### Genome assembly and evaluation

To construct the chromosome-scale assembly of the *V. radiata* genome, we integrated datasets from stLFR reads and Hi-C sequences (Table S3). This initial genome assembly was generated from a total of 501,995,196 reads totaling 121.48 Gb of stLFR sequencing data (225.12 × coverage based on the estimated genome size obtained from k-mer analysis), resulting in a draft genome of 464.02 Mb with 6,685 scaffolds. The longest scaffold was 17.68 Mb, and the N50 length was 4.64 Mb. Additionally, we used 50.32 Gb of Hi-C data to generate chromosome-level scaffolds. The preliminary assembly was scaffolded using the long-range Hi-C technique, resulting in a final assembly containing 468.08 Mb in 5,834 scaffolds with an N50 length of 40.75 Mb (Table S2 and Fig. S1). We estimated the size of the *V. radiata* KUML4 genome based on the k-mer frequency histogram derived from short-read stLFR sequences (Fig. S2). The assembly size (468.08 Mb; Fig. 1) corresponded well with the analysis of k-mer distribution of PE reads (445.98 Mb). Our assembly size was about 2.35% smaller than the estimated size of the Vrad_JL7 genome (479.35 Mb; Table 1) (*Liu et al., 2022*).

The final assembly was anchored into 11 pseudochromosomes larger than 25 Mb, corresponding to the haploid chromosome number in *V. radiata* (2n = 2x = 22). These

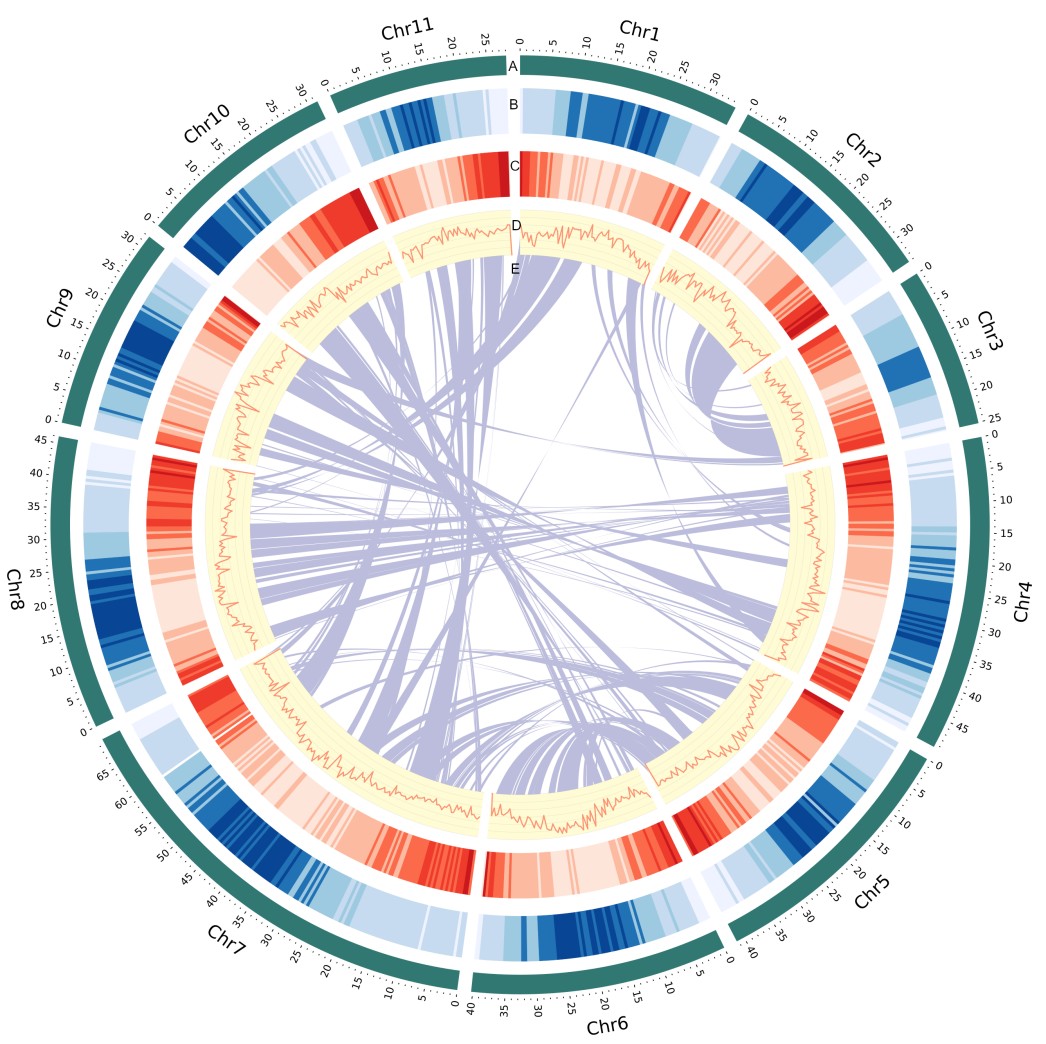

**Figure 1** **Genomic landscape of *V. radiata* chromosomes.** (A) A physical map of 11 pseudomolecules (chromosomes) numbered according to *Liu et al. (2022)*. (B) Repeat density represented by proportion of genomic regions covered by repetitive sequences in 500 kb windows. (C) Gene density represented by number of genes in 500 kb windows. (D) GC content represented by percentage of G + C bases in 500 kb windows. (E) Syntenic blocks in the genome are illustrated by connected lines.

pseudomolecules covered 432.35 Mb, or 96.94% of the estimated genome size and were named in accordance with *Liu et al. (2022)*. To evaluate the quality and accuracy of the *V. radiata* KUML4 assembly, we first mapped the stLFR reads back to the genome assembly and obtained a mapping rate of 98.09%. Second, we evaluated the assembly using 1,614 Benchmarking Universal Single Copy Orthologs (BUSCO) genes from Embryophyta to assess the completeness of the conserved orthologs in the *V. radiata* genome. Our assembly demonstrated high levels of completeness, with proportions of complete (C), complete and single-copy (S), complete and duplicated (D), fragmented (F), and missing (M) genes as follows: C: 98.3% [S: 95.0%, D: 3.1%], F: 0.8%, M: 1.0% (Table S2). Finally, the long terminal repeat (LTR) assembly index (LAI) was employed to assess the completeness of

**Table 1  Assembly statistical comparison of the KUML4, Vrad_JL7, and VC1973A v2 genomes.**

| Genomic feature | KUML4 | Vrad_JL7 | VC1973A v2 |
|---|---|---|---|
| Total assembly size, Mb | 468.08 | 475.35 | 475.7 |
| No. contigs | 5,834 | 632 | 1,511 |
| Largest contig, Mb | 17.68 | 30.20 | 12.73 |
| Contig N50, Mb | 4.64 | 10.34 | 2.8 |
| Scaffold N50, Mb | 40.75 | 43.79 | 47.1 |
| Percentage anchored to chromosomes | 93.52 | 98.72 | 89.92 |
| No. gaps | 15,536 | 259 | 1,047 |
| Length of gaps, Mb | 14.45 | 0.11 | 1.81 |
| GC content, % | 33.27 | 33.45 | 33.27 |
| Complete BUSCOs (genome), % | 98.0 | 98.02 | 91.36 |
| Complete BUSCOs (protein), % | 94.1 | 96.9 | 80.01 |
| LAI | 11.57 | 15.67 | 14.65 |
| Intact LTR-RTs | 1,525 | 2,725 | 2,458 |
| Repetitive sequences, % | 54.76 | 53.45 | 52.79 |
| Protein-coding genes | 27,667 | 40,125 | 30,958 |

the assembled LTR as an indicator of the quality of the whole genome assembly. The LAI score of the assembly was 11.57 (Table 1), meeting the reference genome level according to developer classification standards (*Ou, Chen & Jiang, 2018*).

## Gene annotation

A combination of several types of evidence, including ab initio prediction, RNA-seq supported evidence, and protein homology evidence, was integrated into the genome annotation pipeline to obtain reliable gene predictions. The genome annotation contained 28,593 predicted gene models, of which 27,667 (96.76%) were protein-coding genes. The BUSCO completeness for the predicted genes was 94.1%. The average gene length was 2,793 bp with 4.65 exons per gene, and the average exon and intron length were 236 bp and 464.93 bp, respectively (Table 2). We observed that average GC content of the *V. radiata* genome assembly was 33.27% (Table S2), which was close to the average GC content in introns (32.16%), whereas the average GC content in exons was higher at 44.50% (Table 2). A total of 18,334 predicted genes had been assigned gene ontology (GO) terms. The top three most prevalent GO terms associated with biological process were protein phosphorylation, regulation of DNA-templated transcription, and transmembrane transport, while the most prevalent terms associated with the cellular component were membrane followed by nucleus, and cytoplasm. The largest categories of genes assigned to molecular function were ATP binding followed by metal ion binding, and DNA binding (Fig. S3). Of 28,593 predicted gene models, 96.76%, 70.52%, and 64.12%, could be identified functionally using the NCBI nonredundant (NR) protein, SwissProt and GO databases, respectively (Table S4). For the non-coding RNAs, we also identified 28,017 microRNAs, 765 transfer RNAs, 483 ribosomal RNAs and 12,497 small nuclear RNAs in the *V. radiata* genome (Table S5).

**Table 2  Annotation statistics for *V. radiata* KUML4 genome.**

| | |
|---|---:|
| Number of predicted gene models | 28,593 |
| Number of protein-coding genes | 27,667 |
| Total gene length (Mb) | 79.88 |
| Average gene size (bp) | 2,793.58 |
| Average number of exons/gene | 4.65 |
| Total exon length (Mb) | 31.47 |
| Average exon length (bp) | 236.74 |
| GC content of exons (%) | 44.50 |
| Average number of introns/gene | 3.48 |
| Total intron length (Mb) | 46.27 |
| Average intron length (bp) | 464.93 |
| GC content of introns (%) | 32.16 |

## Identification of repetitive elements

The *V. radiata* KUML4 genome assembly contained a total of 253.85 Mb of repetitive elements, accounting 54.76% of the assembly (Table 3). These repeat elements comprised 25.72 Mb (10.12%) of DNA transposons, 8.04 Mb (3.17%) of simple sequence repeats, 119.09 Mb (46.92%) of retrotransposons, 35.49 Mb (13.98%) of tandem repeats, 18.53 Mb (7.30%) of miniature inverted-repeat transposable elements (MITEs), and 1.60 Mb (0.63%) of helitron elements. Retrotransposons constituted the majority of known repeats, comprising nearly half of the total repeat content in the *V. radiata* KUML4 genome. Within the retrotransposon class, long terminal repeat (LTR) retrotransposons were the most frequently detected, accounting for 46.23% of the repetitive sequences. The most abundant LTR superfamilies, *Gypsy* and *Copia*, accounted for 15.65% and 9.65% of the mungbean genome, respectively (Table 3), which differed from the 16.65% and 13.77% in the Vrad_JL7 genome (*Liu et al., 2022*).

## Comparative genomic analysis

To explore the evolutionary relationships among the three mungbean cultivars (KUML4, Vrad_JL7, and VC1973A v2) and other plant species, the gene sets from the five *Vigna* species (*V. radiata*, *V. mungo*, *V. reflexo-pilosa*, *V. unguiculata*, and *V. angularis*), five other legumes (*P. vulgaris*, *M. truncatula*, *G. max*, *C. arietinum*, and *A. duranensis*), four other eudicots (*A. thaliana*, *C. sativus*, *C. melo*, *C. lanatus*), as well as one monocot (*O. sativa*) were used for gene family clustering. Of the total 670,738 proteins in 17 genomes from 15 species, 641,506 proteins (95.6%) were clustered into 36,504 orthologous groups. Among the 27,625 protein-coding genes in KUML4 that had orthologs in the other species analyzed, only 256 (0.93%) were specific to KUML4, while 575 (2.08%) were shared among legume species (Fig. 2A). Comparison of gene families among the three mungbean genomes revealed that 238, 1,150, and 308 orthologous groups were unique to KUML4, Vrad_JL7 and VC1973A v2, respectively, while 18,152 orthologous groups were shared among KUML4 and the two previously published mungbean genomes (Vrad_JL7 and VC1973A v2; Fig. 2B). Sequence information from single-copy orthologous genes was

**Table 3  Repeat elements in the *V. radiata* KUML4 genome assembly.**

| Types of repeats | Bases (Mb) | % of the assembly | % of total repeats |
|---|---|---|---|
| **DNA transposons:** | | | |
| En-Spm | 13.01 | 2.81 | 5.12 |
| MULE-MuDR | 4.06 | 0.88 | 1.60 |
| DNA: Others | 8.65 | 1.87 | 3.40 |
| **Retrotransposons:** | | | |
| LINE | 1.75 | 0.38 | 0.69 |
| SINE | 0.0015 | 0.00 | 0.00 |
| LTR: *Copia* | 44.63 | 9.65 | 17.58 |
| LTR: *Gypsy* | 72.34 | 15.65 | 28.50 |
| LTR: Others | 0.37 | 0.08 | 0.15 |
| **Simple sequence repeats:** | 8.04 | 1.74 | 3.17 |
| **Tandem repeats:** | 35.49 | 7.58 | 13.98 |
| **MITES:** | 18.53 | 3.96 | 7.30 |
| **Helitrons:** | 1.60 | 0.34 | 0.63 |
| **Others:** | 45.38 | 9.82 | 17.88 |
| **Total** | 253.85 | 57.46 | |

used to construct a maximum-likelihood tree with *O. sativa* as an outgroup, and the divergence time was estimated based on the topology and branch length. The phylogenetic analysis revealed that the three *V. radiata* cultivars (KUML4, Vrad_JL7 and VC1973A v2) and *V. mungo* were closest and diverged approximately 4.17 million years ago (MYA). The divergence of KUML4 from Vrad_JL7 occurred around 1.38 MYA. The ancestor of *V. radiata* and *V. mungo* formed a sister clade to the ancestor of *V. angularis* and *V. reflexo-pilosa*. The estimated divergence time between the two clades was about 7 MYA (Fig. 2C). Additionally, our analysis of gene family expansion and contraction across the three mungbean cultivars and other plant species identified 2,764 significant gene families. A total of 39 significantly expanded gene families, and 932 significantly contracted families were identified in KUML4 (Fig. 2C).

Functional categories indicated that a large number of expanded gene families were associated with secondary metabolite biosynthesis, including berberine bridge enzyme-like 13, isoflavone 2′-hydroxylase, UDP-glycosyltransferase 73D1. In addition, they were enriched in the defense responses, and signal transduction related to plant hormones like ethylene and cytokinin, including aspartic proteinase CDR1-like, GDSL esterase/lipase EXL3, ethylene-responsive transcription factor ERF098-like, zeatin O-glucosyltransferase-like, and cytokinin riboside 5′-monophosphate phosphoribohydrolase (Table S6). The contracted gene families were mainly involved in the cysteine-rich receptor-like protein kinase 10, proline-rich receptor-like protein kinase PERK4, shaggy-related protein kinase epsilon isoform X1, and MDIS1-interacting receptor-like kinase (Table S6).

We also explored genome synteny among *V. radiata* KUML4, *V. radiata* JL7, and *V. mungo*. Our genome assembly of KUML4 reveals a high level of synteny with the Vrad_JL7 genome, as illustrated by the large syntenic blocks in Fig. 3. Although some

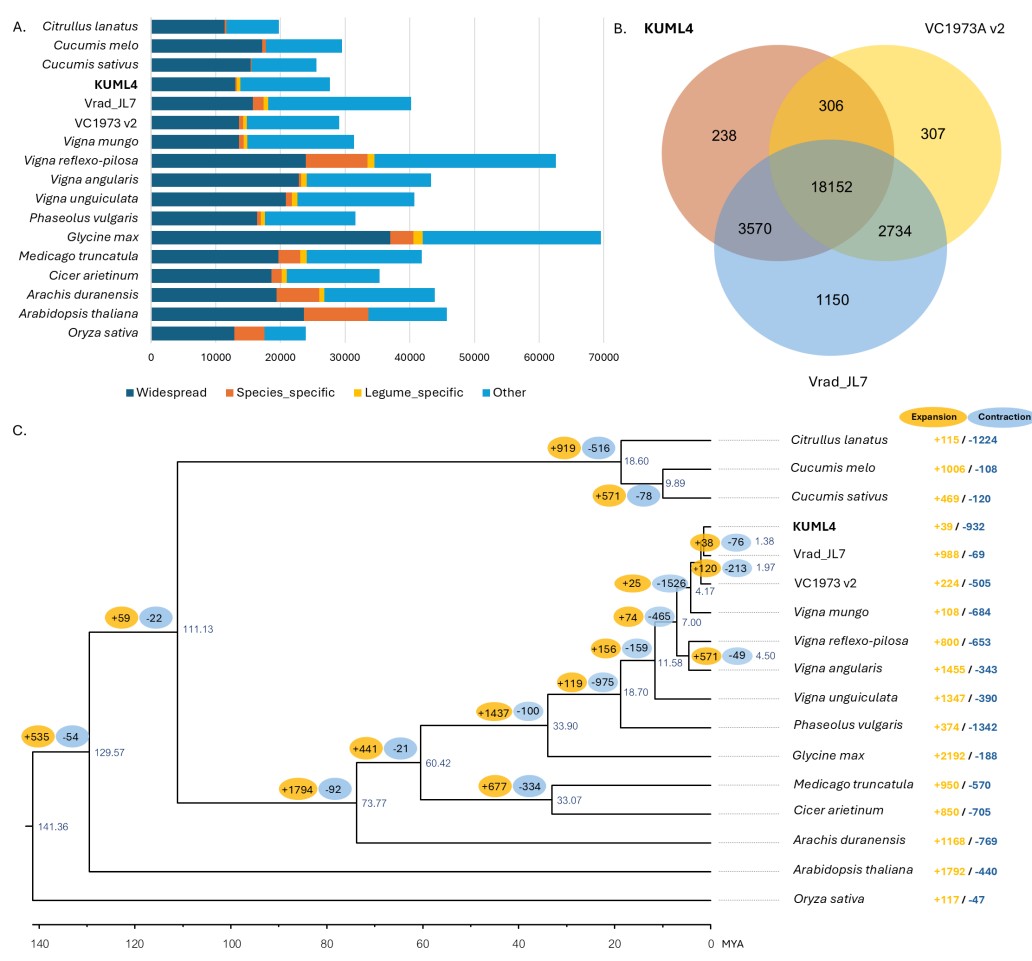

**Figure 2** Comparative genomics of mungbean and other plant species. (A) Bar charts display the number of proteins that were widespread, legume-specific and species-specific (having no detectable orthologues in other species). (B) Venn diagrams displaying clusters of shared and unique orthologous gene families in three mugbean genomes (KUML4, Vrad_JL7, and VC1973A v2). (C) Inferred phylogenetic tree of mungbean and other plant species based on protein sequences of single-copy orthologous genes. Numbers at each node represent the estimated divergence time of each node in million years ago (MYA). The branch labels in yellow and blue represent the significantly expanded and contracted gene families, respectively, of each node. The right column shows significantly expanded and contracted gene families of individual species.

structural variations were detected, including inverted regions, these were observed on chromosomes 1, 2, 3, 6, 9, 10, and 11 between the KUML4 and Vrad_JL7 genomes.

# DISCUSSION

The mungbean is one of the most important edible legume crops, widely cultivated in many Asian countries (*Hou et al., 2019*). It has significant health and economic benefits and also plays a significant role in climate resilience and increasing food security (*Assefa, Dinku & Jemal, 2022*). Genome assemblies of the mungbean are fundamental to understanding its genetic makeup, which is essential for genetic improvement and superior cultivar
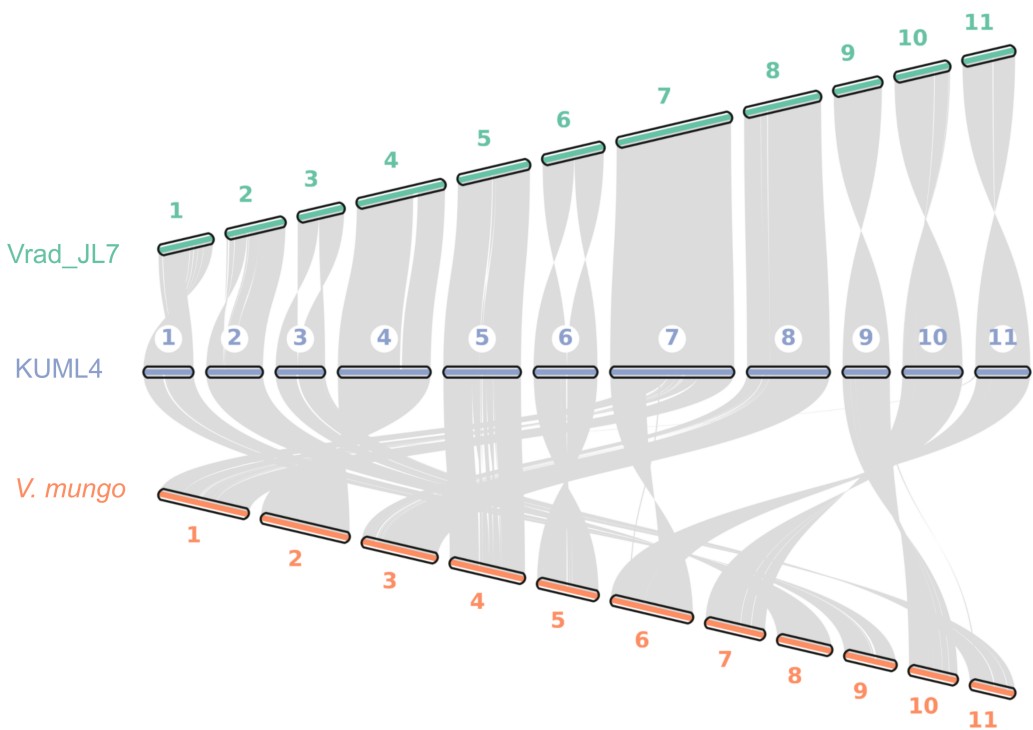

**Figure 3  Chromosome collinearity between *V. radiata*. cultivar KML4, *V. radiata* cultivar Vrad_JL7, and *V. mungo*.**

development. In the present study, we sequenced and assembled the reference genome of the mungbean cultivar KUML4, a new cultivar grown for dry seed in Thailand. By combining the stLFR technique with the high-throughput chromosome conformation capture (Hi-C) technique, we achieved a chromosome-scale assembly for the cultivar KUML4. The stLFR technology enables sequencing of data from long DNA molecules by adding the same barcode sequence to sub-fragments of the original long DNA molecule (*Wang et al., 2019*). The Hi-C approach facilitates the analysis of chromosomal interactions through chromosome conformation capture. Hi-C data can provide long-range linkage information up to tens of megabases that could be used to generate chromosome-scale scaffolds (*Burton et al., 2013*; *Ghurye et al., 2019*; *Marie-Nelly et al., 2014*). Our final assembly comprised 11 pseudochromosomes covering 432.35 Mb, accounting for 96.94% of the 445.98 Mb estimated genome size, with an N50 length of 40.75 Mb. Compared with three previously reported munngbean genome assemblies—VC1973A (421 Mb) (*Kang et al., 2014*), the improved version of VC1973A (476 Mb) (*Ha et al., 2021*), and Vrad_JL7 (479 Mb) (*Liu et al., 2022*)—our mungbean assembly (KUML4) is longer in size than the initial draft of the VC1973A genome and shorter than the improved version of VC1973A and the Vrad_JL7 genomes. The completeness of the gene space measured by BUSCO in our assembly is higher than that of the two VC1973A assemblies (*Ha et al., 2021*; *Kang et al., 2014*) and comparable to the assembly of Vrad_JL7 (98.3% for our assembly and 98.02% for Vrad_JL7

(*Liu et al., 2022*), suggesting that both the KUML4 and Vrad_JL7 assemblies achieve a high level of completeness.

We also performed a synteny analysis between our assembled genome, Vrad_JL7, and *V. mungo*. The genome comparison revealed high collinearity between the chromosomes of mungbean KUML4 and Vrad_JL7 (Fig. 3). However, some genomic regions in KUML4, Vrad_JL7, and *V. mungo* appeared inverted in the comparison, particularly on chromosomes 1, 2, 3, 6, 9, 10, and 11. *Liu et al. (2022)* suggested that these inversions observed in Vrad_JL7, as well as in cowpea and adzuki bean, may have resulted from misassembly in Vrad_JL7. The study also suggested that the density distribution of genes and transposable elements indicated that certain segments on chromosomes 1, 2, 3, 6, and 10 might need to be reversed in orientation.

The KUML4 assembly is not significantly improved compared to the Vrad_JL7 assembly, with several evaluation scores, including contig number, scaffold N50 length, and gap count, showing both lower and higher values (Table 1). According to the LAI-based method for assessing genome assembly quality, our mungbean KUML4 genome assembly shows a slightly lower LAI score than the two previously published mungbean genomes (Vrad_JL7 and VC1973A; Table 1). However, the KUML4 assembly still offers high-quality synteny and gene content, making it a valuable genome for comparative analysis. The LAI scale typically considers scores above 10 as indicative of reference-quality assemblies (*Ou, Chen & Jiang, 2018*). The differences in LAI scores may be attributed to several factors, such as variations in sequencing technology, assembly methods, or data quality. LAI is one of several metrics used to assess genome quality. However, based on contig N50 size, scaffold N50 size, BUSCO scores, and the completeness of the functional annotation (Table 1), our KUML4 genome assembly qualifies as a draft reference-quality genome. Its good contiguity and accuracy provide a valuable reference for future genetic research on other legume species, and the newly assembled KUML4 genome contributes significantly to the legume genome resource bank, offering a novel genomic resource for future mungbean improvement efforts. In addition, the number of predicted genes (28,593) in our assembly is higher than that predicted in the initial draft genome assembly of mungbean (*Kang et al., 2014*), but lower than that of the latest assembly (*Liu et al., 2022*). We annotated 27,667 protein-coding genes in our assembled genome, while the published Vrad_JL7 genome annotated 40,125 genes. The discrepancy in gene numbers between our KUML4 assembly and Vrad_JL7 can be attributed to several factors. Among the 40,125 protein-coding genes in the Vrad_JL7 genome, approximately 32,000 were functionally annotated, while the remaining ~7,400 genes had minimal hits in the NCBI nr database during BLAST searches. According to orthogroup analysis of the three *V. radiata* genomes (KUML4, Vrad_JL7, and VC1973A v2), 1,150 orthogroups consisting of 3,437 genes were unique to Vrad_JL7 (Fig. 2B). Of these, ~1,900 genes were not functionally annotated, and ~2,700 of the 7,400 genes were not assigned to any orthogroup. Therefore, approximately 4,600 to 7,400 genes may be unique to Vrad_JL7. Additionally, *Liu et al. (2022)* noted that some gene models in Vrad_JL7 could be improved, highlighting that some gene models were short (less than 200 nt), while others were partial, fragmented, repetitive, or chimeric. The complexity of genetic variation among mungbean cultivars plays a significant role.

The different agronomic traits of KUML4 and Vrad_JL7 cultivars likely contribute to differences in genome content, including gene number and structural variations, with each cultivar having cultivar-specific genes based on its breeding history. Furthermore, differences in sequencing technologies and assembly pipelines compared to previous studies may contribute to discrepancies in gene numbers and structural variations. The use of stLFR sequencing and Hi-C for genome assembly in KUML4 differs from the long-read sequencing employed in Vrad_JL7. For instance, variations in read lengths, coverage depth, and assembly algorithms can influence the detection of repetitive sequences, gene fragmentation, and overall assembly completeness.

However, the BUSCO completeness for the predicted genes assessment revealed 94.1% in our assembly, which is close to the 96.9% reported in a previous study (*Liu et al., 2022*), indicating the high completeness of gene annotation. Compared with other sequenced legume genomes, the number of predicted genes in our mungbean genome is lower than that of black gram (*Pootakham et al., 2021*), adzuki bean (*Yang et al., 2015*), soybean (*Schmutz et al., 2010*) and pigeonpea (*Varshney et al., 2012*), and higher than that of common bean (*Schmutz et al., 2014*), but similar to chickpea (*Varshney et al., 2013*). The proportion of repetitive sequences identified in our genome assembly (54.76%) is slightly higher than the percentage reported for mungbean Vrad_JL7 (53.45%) in *Liu et al. (2022)*. LTR elements, including Gypsy and Copia elements were the predominant classes in our mungbean assembly. Compared to Copia elements (9.65%), Gypsy elements (15.65%) occupied relatively larger proportions of genomic sequence in mungbean, which is consistent with earlier reports about other *Vigna* species (*Guan et al., 2022*; *Pootakham et al., 2021*; *Pootakham et al., 2023*).

To explore the genetic distinctiveness of mungbean, we focused on expanded orthologous gene families and their gene functions. Changes in genetic makeup are potentially linked to unique phenotypic and functional changes. By examining expanded gene families in mungbean, we found an enrichment in genes associated with secondary metabolite biosynthesis, defense responses, and signal transduction related to plant hormones like ethylene and cytokinin. Notably, we observed that several genes associated with the auxin signaling pathway, including auxin-induced protein X10A-like, were expanded. This pathway plays a key role in regulating cell elongation and division, which directly influence seed size and yield. Additionally, genes involved in cytokinin response pathways, crucial for promoting cell proliferation and grain filling, were also enriched. These gene family expansions likely contribute to KUML4's high yield and large seed size by supporting growth-related processes. A phylogenetic analysis based on sequence information from single-copy orthologous genes revealed that mungbean and black gram were the most closely related, and the ancestor of mungbean and black gram formed a sister clade to the ancestor of créole bean and azuki bean, consistent with the results of previously published phylogenetic analyses (*Doi et al., 2002*; *Pootakham et al., 2021*).

The KUML4 genome assembly provides valuable insights due to its distinct genetic background and unique agronomic traits. These traits set KUML4 apart from other *V. radiata* cultivars, making this genome a crucial addition to the understanding of the species' genetic diversity. Developing mungbean cultivars with desirable agronomic traits,

such as enhanced productivity and tolerance to biotic and abiotic stresses, has long been a key goal in breeding programs. We believe that having a chromosome-scale reference genome for KUML4 will significantly enhance our understanding of mungbean biology, thereby greatly aiding molecular breeding efforts to produce improved mungbean cultivars.

## CONCLUSIONS

Mungbean genome assemblies are key to understanding the genetic basis of its ability to adapt to various environments and produce high nutritional content. Utilizing the stLFR technique together with high-throughput chromosome conformation capture (Hi-C) technique, we achieved a chromosome-scale assembly of the mungbean cultivar KUML4. The assembly encompassed 11 pseudochromosomes with a total length of 468.08 Mb and a scaffold N50 of 40.75 Mb. The genomic resource developed here will provide valuable information for future mungbean genetic improvement. Furthermore, it serves as a valuable addition to ongoing comparative genomics research within legume species.

### Funding

This project was funded by the National Research Council of Thailand (NRCT) (grant no. N42A650274) and Kasetsart University Research and Development Institute. The funders had no role in study design, data collection and analysis, decision to publish, or preparation of the manuscript.

### Grant Disclosures

The following grant information was disclosed by the authors:
National Research Council of Thailand (NRCT): N42A650274.
Kasetsart University Research and Development Institute.

### Competing Interests

The authors declare there are no competing interests.

### Author Contributions

- Supaporn Khanbo conceived and designed the experiments, prepared figures and/or tables, authored or reviewed drafts of the article, and approved the final draft.
- Poompat Phadphon analyzed the data, prepared figures and/or tables, and approved the final draft.
- Chaiwat Naktang analyzed the data, prepared figures and/or tables, and approved the final draft.
- Duangjai Sangsrakru performed the experiments, prepared figures and/or tables, and approved the final draft.
- Pitchaporn Waiyamitra performed the experiments, prepared figures and/or tables, and approved the final draft.

- Nattapol Narong performed the experiments, prepared figures and/or tables, and approved the final draft.
- Chutintorn Yundaeng analyzed the data, prepared figures and/or tables, and approved the final draft.
- Sithichoke Tangphatsornruang conceived and designed the experiments, authored or reviewed drafts of the article, and approved the final draft.
- Kularb Laosatit conceived and designed the experiments, authored or reviewed drafts of the article, and approved the final draft.
- Prakit Somta conceived and designed the experiments, authored or reviewed drafts of the article, and approved the final draft.
- Wirulda Pootakham conceived and designed the experiments, authored or reviewed drafts of the article, and approved the final draft.

## Data Availability

The genome assembly data is available at NCBI: JAVCZA000000000, GCA_040583985. The raw reads are available at NCBI SRA: SRR29729411.

The genome annotation of mungbean is available at Figshare: Sonthirod, Chutima (2024). *Vigna radiata* annotation files. figshare. Dataset. https://doi.org/10.6084/m9.figshare.27094231.v1

## Supplemental Information

Supplemental information for this article can be found online at http://dx.doi.org/10.7717/peerj.18771#supplemental-information.

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
