# Peer review of "A chromosome-scale genome assembly of mungbean (Vigna radiata)"

_PeerJ, doi:10.7717/peerj.18771_

## Round 0.1 · original submission · Major Revisions

There are at least two reported mung bean genome assemblies. Your current manuscript shows a potential to report an improved genome assembly of V. radiata and its effect on comparative genomics. However, the authors do not provide convincing evidence that an assembled KUML4 genome alone can serve as a single representation of the V. radiata genome for comparative genomics. Both reviewers have raised concerns on your assembly. Particularly, Reviewer 2 made comments on comparing the assembled KUML4 genome with previous V. radiata genomes. Please also consider the complexity of genetic variation among plant samples (i.e., different agronomic traits among reference mung beans, Liu’s, Ha’s, and KUML4) and technological limitations on how the authors’ genome assembly was done.

Please provide a point-by-point response to the reviewers’ comments. In addition, please specify “short-read DNA sequences” in line 129; double-check details of transcriptomes in line 152. The authors listed 6 numbers; how about yield and seed size-related genes regarding the gene expansion and contraction (e.g., line 272) and signal pathways (e.g., line 279)?; what happens if you set at p < 0.05 (line 273)?; rephrase line 305. Your genome is longer in size than Kang et al. and shorter than Liu et al., Ha et al.; provide details of the identification/version/reference of program/software/package or a date if the outsourcing of bioinformatics has been done; please specify “both assemblies” in line 309.

Reviewer 1 ·

Basic reporting

no comment

Experimental design

no comment

Validity of the findings

no comment

Additional comments

Whole genome sequences of different cultivars are important for the discovery of genes and molecular markers associated with diverse agronomic traits. In this study, the authors assembled a high-quality chromosome-level genome of the mungbean, KUML4, based on stLFR sequencing combined with Hi-C technology. This high-quality genome assembly serves as a valuable resource for accelerating the biological discovery and molecular breeding of mungbean. But there are some problems with this manuscript. For the moment, it is not proper to public with this version.
1. The raw sequencing data generated in this study should be publicly accessible. The authors could share these data at NGDC (https://ngdc.cncb.ac.cn/) or NCBI (https://www.ncbi.nlm.nih.gov/).
2. The assembled genome sequence and gene annotations in this study could be shared at GenBank (https://www.ncbi.nlm.nih.gov/genbank/) or figshare (https://figshare.com/).
3. Table 1 shows the same BUSCO score of genome assemblies generated from “stLFR sequencing” and “stLFR sequencing + HiC”, indicating that this software may be not suitable for evaluating this genome assembly. The author could use the LTR Assembly Index (LAI) to assess the genome completeness.
4. In the reference section, the journal names are inconsistent. For example, both “Nat Commun” and “Nature Communications” exist. The authors need to check all the references carefully.

Reviewer 2 ·

Basic reporting

The manuscript is about the generation of a high-quality, chromosome-scale genome assembly of mungbean (Vigna radiata), emphasizing its importance as a leguminous crop in Asia. Considering that there are already available genomes for the species the novelty is employing single-tube long-fragment reads (stLFR) sequencing technology combined with chromatin contact mapping (Hi-C). Once the assembly is done, authors should look at the quality of assembly by comparing it with already reported assemblies, which is largely missing. Any new assembly for a species provides two purposes: Firstly, It is an addition to the existing information about the species genome as an additional resource. Secondly, a comparison with the existing genome provides whether there is an improvement on the existing genome. Manuscript although promising, falters on both aspects. My recommendation is to revise it to include both of these aspects. Following are my detailed comments to help authors to improve the manuscript.

Experimental design

The methodology section requires improvement for clarity and reproducibility. The versions of tools used, such as HiRise and BWA, are not specified, and the parameters applied in the analysis are not detailed. Additionally, the manuscript lacks uniformity in reporting tool names (e.g., bwa vs. BWA), which should be standardized. The depth and quality of raw data, including reads remaining after filtration for both stLFR and Hi-C, should be provided in a supplementary table. The final step of manual curation in Hi-C analysis, which is a standard practice, is not mentioned and should be included. Comparison with existing genomes is missing.

Validity of the findings

Validation of the data is missing.
1. The coverage of the assembled genome is calculated from the final assembly size, which is not the most accurate approach. The authors should perform a k-mer based genome size estimation and then calculate the coverage. However, k-mer based methods can underestimate genome size, so the reported literature genome size or flow cytometry should be considered for calculating the coverage of the final assembly (Arumuganathan et al., 1991).
2. The section on the identification of repetitive elements only discusses one pipeline for characterization, missing other elements like MITEs, Helitrons, and Tandem Repeats, which should also be characterized. Intact LTR elements and the LAI index, which are also standard assessments for genome quality now days, are not mentioned.
3. Comparison with existing genomes is missing. The references for all species used in the comparative analysis are missing. A comprehensive comparative analysis with the already published Vigna radiata genomes, particularly JL7, should be included to demonstrate the value added by the present genome assembly. The number of predicted genes is significantly lower (~28,000) compared to JL7 (~40,000), which is unusual given the similar final assembly sizes. This discrepancy requires further investigation and clarification.
4. The number of gaps and N’s incorporated in the primary and final assembly should be reported.
5. The language and formatting need to be looked at and need a thorough revision.

Additional comments

Specific Suggestions:
- The manuscript should include detailed tool versions and parameters used in the analysis for reproducibility.
- Raw data statistics before and after filtering for both stLFR and Hi-C, including read depth and quality, should be provided in a supplementary table.
- The methodology for genome size estimation should be revised to include k-mer analysis and consideration of literature or flow cytometry data.
- The identification of repetitive elements should include a broader range of methods to capture all types of repetitive sequences.
- A comprehensive comparative analysis with existing Vigna radiata genomes JL7 should be added to demonstrate the novelty and improvements of the current assembly if any.
- The discrepancy in gene prediction numbers should be thoroughly investigated and explained.
- The number of gaps and N’s incorporated in the primary and final assembly should be reported.
- Line 108 should be corrected by italicizing HindIII.

---

## Round 0.2 · Major Revisions

Thank you for your revised manuscript. We have now received two reviewer reports. It becomes clear from their reports that the present study is of potential interest for publication. Nevertheless, the reviewer #2 voices concerns regarding analyses and presentation, and it is clear that an additional revision will be required before/if this study is to be published. Please provide a point-by-point response to the reviewer #2’ comments. From my own reading of your manuscript: Any further insight the authors can bring to the genes related to yield and seed size would be a nice addition to the manuscript. The authors can consider 2-3 sentences on what happens on those genes (Similar to the authors’ answer to my previous comment regarding the gene expansion and contraction and signal pathways.)

Reviewer 1 ·

Basic reporting

no comment

Experimental design

no comment

Validity of the findings

no comment

Additional comments

The authors have satisfactorily addressed my previous concerns and significantly improved the manuscript. I have no further comments on the revised manuscript.

Reviewer 2 ·

Basic reporting

The revised manuscript has addressed the initial concerns, but the additional data provided has raised many new concerns. However, all these concerns can be addressed, although this means that the manuscript has to go under another round of revision, I hope the authors will understand the need for it.

Experimental design

A thorough review shows that the KUML4 genome assembly does not exhibit significant improvements over the previously published Vrad_JL7 genome, mainly based on several completeness and quality metrics. Below are my observations and suggestions to improve the manuscript:
1. Given the following observations, describing KUML4 as a draft rather than a high-quality genome would be more accurate. I recommend revising the title and any instances in the manuscript that refer to KUML4 as an improved or high-quality genome.
Assembly Quality Comparison: The KUML4 genome does not currently demonstrate improved contiguity or completeness compared to Vrad_JL7. Specifically:
a) Contig Number: KUML4 has a significantly higher contig count (5834) compared to Vrad_JL7 (632) and VC1973A v2 (1511), indicating a less contiguous assembly.
b) Scaffold N50: The N50 of KUML4 is 40.75 Mb, which is lower than Vrad_JL7’s 43.79 Mb.
c) Gap Count and Length: KUML4 contains 15,536 gaps with a total gap length of 14.45 Mb, which is substantially higher than Vrad_JL7’s 259 gaps and 0.11 Mb gap length, indicating more unresolved regions in KUML4.
d) LAI Score: KUML4’s LAI score of 11.57 is lower than Vrad_JL7’s 15.67, suggesting a lower assembly quality for repetitive regions.
e) Intact LTR-RTs: KUML4 includes 1525 intact LTR-RTs, fewer than 2725 observed in Vrad_JL7.
2. Claims of misassembly in Vrad_JL7: Authors have suggested that KUML4 corrects potential misassemblies in Vrad_JL7. This is an interesting point; however, these statements remain speculative without supporting evidence. I suggest providing a detailed bioinformatics analysis demonstrating specific regions or miss-assembly types in Vrad_JL7 to strengthen this claim. This will add value to the KUML4 genome, improving the previously reported genomes. I highly recommend performing some in-depth analysis to resolve this miss-assembly. To explore this, consider performing detailed structural comparisons using alignment-based methods to detect discrepancies between KUML4 and Vrad_JL7. Additionally, analysis focusing on complex genomic regions, such as telomeres and centromeres, may reveal structural inconsistencies or assembly gaps.
a) Authors can map against existing long-read sequencing, scaffolding, and Hi-C assembly data. This will help clarify whether the observed differences represent true miss-assemblies or reflect the natural variation.
b) Whole-Genome Alignment and Structural Variants: Authors can conduct an in-depth whole-genome alignment to identify structural variants, such as inversions, translocations, or large insertions/deletions, between KUML4 and Vrad_JL7. This type of analysis can reveal potential misassemblies that might not be visible in more localized comparisons.
c) Read Depth Analysis: A read depth consistency check across both genomes is recommended. Areas with abnormal read depth (very high or low coverage) could indicate misassemblies or collapsed repeat regions.
d) Telomere and Centromere Integrity: Advise a focus on telomere and centromere integrity, as these regions are often challenging to assemble and prone to misassembly. Specialized tools that analyze repeat structures can help assess completeness and correctness.
e) It is needed to be established if the misassembly is in Vrad_JL7 or KUML4.

Validity of the findings

Please see the comments above

Additional comments

Authors need to do some additional analysis. They largely depend on previous reporting when suggesting misassemblies (L 397 to 401 in discussion). Additionally, they are trying to prove that they are better than previous assemblies, although many parameters suggest otherwise. They need to be careful in their writing.

---

## Round 0.3 · Minor Revisions

Thank you for your revision. I have read your revision as well as reviewer comments. I would like to address all as follows: 1) The authors use two different terms, ‘chromatin contact mapping’ and ‘chromosome conformation capturing technique’, for the ‘Hi-C’. I somehow understand why those two have been used, but it is not crystal clear. I have listed several cases, lines 24, 77, 106, 121, 343, 447 per peerj-reviewing-102524-v2.pdf. Consider ‘High-throughput chromosome conformation capture (Hi-C) technique’ throughout the text if this is what the authors intended. Must double-check with sequencing service providers as well if they have any preference; 2) Move Figures 1 and 2 and Table 1 to supplementary datasets. For Table 2, it is my understanding that the authors re-analyze previous assemblies. If correct, clarify what the authors did in the text and the table; 3) Remove ‘new and popular’ from line 79; 4) Double-check JAVCZA000000000 in line 133 and SRR29729411 in line 134 if IDs are correct; 5) Specify the raw reads stored in line 134. Both stLFR and Hi-C?; 6) Rewrite ‘While the Hi-C approach enables the analysis of chromosomal interactions using chromosome conformation capture.’ in line 346. Lastly, the reviewer in review round 2 has provided very good comments throughout review rounds such as deep bioinformatics to previous and current assemblies to evaluate potential false assemblies and the flow cytometry. Since this manuscript presents a different genotype previous assemblies and the new assembly together would either provide triple reference, or lead deeper follow-up studies in genomics later. I wouldn’t request additional experiments.

Reviewer 2 ·

Basic reporting

Basic data generated is fine. Authors have used different sequencing technology compared to previous report. The language written is readable. Structure of the manuscript is straight forward.

Experimental design

Experimental design is straightforward. But the authors are just reporting data. They have not made any attempt to improve on previous datasets reported. Their comparison to the previous assembly shows that they have not improved and are showing lower indices than the previous one. Authors are unwilling to do any additional analysis.

Validity of the findings

This is where I feel authors could do more. A fair comparison indicates lower indices for the current data.

Additional comments

I think the manuscript can be published for the data generated in an additional accession. Besides that manuscript does not show any additional information or improve the assembly in any way.

---

## Round 0.4 · accepted · Accept

Thank you for addressing all comments. I am pleased to accept your manuscript for publication in PeerJ.